# Structural basis for mutually exclusive co-transcriptional nuclear cap-binding complexes with either NELF-E or ARS2

Wiebke Manuela Schulze[1,2] & Stephen Cusack [1]

Pol II transcribes diverse classes of RNAs that need to be directed into the appropriate nuclear maturation pathway. All nascent Pol II transcripts are 5′-capped and the cap is immediately sequestered by the nuclear cap-binding complex (CBC). Mutually exclusive interactions of CBC with different partner proteins have been implicated in transcript fate determination. Here, we characterise the direct interactions between CBC and NELF-E, a subunit of the negative elongation factor complex, ARS2 and PHAX. Our biochemical and crystal structure results show that the homologous C-terminal peptides of NELF-E and ARS2 bind identically to CBC and in each case the affinity is enhanced when CBC is bound to a cap analogue. Furthermore, whereas PHAX forms a complex with CBC and ARS2, NELF-E binding to CBC is incompatible with PHAX binding. We thus define two mutually exclusive complexes CBC–NELF–E and CBC–ARS2–PHAX, which likely act in respectively earlier and later phases of transcription.

[1] European Molecular Biology Laboratory, Grenoble Outstation, 71 Avenue des Martyrs, CS 90181, 38042 Grenoble Cedex 9, France. [2] Ruprecht-Karls University of Heidelberg, Grabengasse 1, 69117 Heidelberg, Germany. Correspondence and requests for materials should be addressed to S.C. (email: cusack@embl.fr)

The nuclear cap-binding complex (CBC) binds tightly to the 5′-cap structure (m[7]GpppN, where N is the first transcribed nucleotide) which is co-transcriptionally added to all nascent Pol II transcripts 25–50 nucleotides after initiation. The cap binds by an induced fit mechanism to the RRM/RNP domain of CBP20, the small subunit of CBC[1–3]. However, high-affinity cap binding requires that CBP20 forms a heterodimer with the CBP80, the large and predominantly helical subunit of CBC[1,2,4]. CBC

bound to capped RNA not only protects the transcript from decapping and subsequent further 5′-3′ degradation, but also acts as a platform for interaction with nuclear factors that determine the fate of the transcript[5,6]. For example, in metazoans, the transport factor PHAX (phosphorylated adaptor for RNA export) binds directly to CBC as well as the capped RNA[7,8] to promote nuclear export of snRNAs[9,10] or the intra-nuclear transport of snoRNAs[11]. For mRNAs, CBC has been shown to enhance cap-

**Fig. 1** The interaction of CBC with ARS2 is enhanced in the presence of m[7]GTP. **a** Schematic diagram of the domain structure of ARS2 based on ref. [26]. **b** Gel filtration chromatogram and Coomassie-stained SDS-PAGE of CBC-ARS2 in the presence and absence of m[7]GTP. Purified recombinant CBC and ARS2[147–871] were mixed with (orange) or without (blue) 500 µM m[7]GTP and subjected to gel filtration. Their elution profiles were overlaid (left) and single fractions analysed by SDS-PAGE (middle and right). Purified recombinant CBC and C-terminally truncated ARS2[147–845] were mixed with 500 µM m[7]GTP and analysed in a similar way (bottom left). **c** ITC data and curve fit to derive the affinity of different ARS2 constructs for CBC. CBC or m[7]GTP-CBC in the sample cell was titrated by ARS2. In each panel, the upper graph shows the raw data and the lower graph shows the ligand concentration dependence of the heat released upon binding after normalisation. $K_D$ values and standard deviation represent the average from at least two independent experiments. **d** Sequence alignment of the C-terminus of ARS2. Conserved residues are highlighted in red. Figure made with ESPript[54]

**Table 1 Summary of dissociation constants measured by ITC**

| Interaction analysed | $K_D$ (μM) | Number of ITC experiments |
| --- | --- | --- |
| CBC + ARS2 827–871 | 12.2 ± 3.5 | 2 |
| m⁷GTP-CBC + ARS2 827–871 | 1.0 ± 0.2 | 4 |
| m⁷GTP-CBC + ARS2 763–871 | 1.0 ± 0.3 | 2 |
| m⁷GTP-CBC + ARS2 763–845 | n.d. | 2 |
| m⁷GTP-CBC + ARS2 845–871 | 1.1 ± 0.16 | 2 |
| m⁷GTP-CBC + ARS2 827–871 F871D | 1.8 ± 0.26 | 2 |
| m⁷GTP-CBC + ARS2 827–871 R854A Y859A | n.d. | 2 |
| CBC + PHAX | 0.3 ± 0.16 | 4 |
| m⁷GTP-CBC + PHAX | 0.13 ± 0.05 | 2 |
| CBC + PHAX 103–327 | 0.35 ± 0.071 | 2 |
| CBC + PHAX 103–308 | 0.6 ± 0.14 | 2 |
| CBC + PHAX 103–294 | 0.85 ± 0.21 | 2 |
| CBC + PHAX 120–308 | No interaction via SEC | - |
| CBC + PHAX 103–264 | No interaction via SEC | - |
| CBC + NELF-E 244–380 | 0.4 ± 0.2 | 2 |
| m⁷GTP-CBC + NELF-E 244–380 | 0.05 ± 0.01 | 3 |
| CBC + NELF-E 244–360 | n.d. | 2 |
| m⁷GTP-CBC + NELF-E 360–380 | 3.3 ± 1.1 | 3 |
| m⁷GTP-CBCmutᵃ + ARS2 827–871 | 11.6 ± 1.1 | 2 |
| GTP-CBCmut + PHAX | 0.5 ± 0.14 | 2 |
| m⁷GTP-CBCmut + NELF-E 244–380 | n.d. | 2 |
| m⁷GTP + CBC | 0.126 ± 0.045 | 6 |
| m⁷GTP + CBCmut | 0.095 ± 0.024 | 2 |
| m⁷GTP + CBC (6 mutations) | 70 ± 4 | 2 |
| m⁷GTP + CBC-NELF-E 244–380 | 0.044 ± 0.002 | 3 |
| m⁷GTP + CBC-ARS2 827–871 | 0.018 ± 0.001 | 2 |

*Note*: $K_D$ values and standard deviation represent the average from at least two independent experiments as indicated
ᵃCBCmut combines the single Y50A mutation in CBP20 together with the CBP80 triple mutation Y461A/R610E/H651A

proximal splicing[1,12,13] and 3′-end processing[14] and also facilitates mRNA nuclear export[15,16]. These two examples raise the question of how different classes of transcripts are directed into different nuclear transport and processing pathways. It has been shown that one feature that is used to distinguish snRNAs from pre-mRNAs, for example, is length[17]. For transcripts longer than about 300 nts, a tetramer of hnRNP C is able to displace PHAX from the nascent RNA–CBC complex[18] and thus direct mRNAs into their appropriate nuclear export pathway. This illustrates the notion that at different stages during transcription and nuclear processing, dynamic remodelling of CBC-associated factors can occur through mutually exclusive interactions that eventually determine transcript fate[5]. Here we take a biochemical and structural approach to investigate which binary and ternary complexes CBC can form with three nuclear factors, PHAX, ARS2 (arsenate resistance protein 2), and NELF (negative elongation factor), all of which directly interact with CBC and are involved in CBC-mediated transcript processing.

ARS2 is a highly conserved metazoan protein of around 871–884 residues (depending on isoform) that was shown to be implicated in processing of certain pri-miRNAs in mammalian cells[19,20], *Drosophila*[21], and *Arabidopsis thaliana* (where the protein is known as SERRATE)[22,23]. ARS2 can form a binary complex with CBC and a ternary complex including PHAX as well[24]. Both complexes bind preferentially to a similar set of short capped transcripts including snRNAs, snoRNAs, and mRNAs[24]. ARS2 is thought to bind directly both CBC and its bound capped RNA. It appears to have specificity for stem-loop containing transcripts such as pri-miRNAs, histone pre-mRNAs, and 7SK RNA[20,25,26], although details of the protein–RNA interactions are not known. The most prominent defects upon ARS2 knockdown in mammalian cells are in 3′-end processing of

certain transcripts such as histone mRNAs and snRNAs, as well as in reduced degradation of short-lived transcripts such as Promoter Upstream Transcripts (PROMPTs)[20,24,27]. The important role ARS2 plays in promoting correct processing of replication-dependent histone mRNA as well as in miRNA biogenesis likely explains its requirement for cell-cycle progression[19,26]. The role of ARS2 in histone processing has been partly shown to depend on its direct interaction with FADD-like IL-1beta-converting enzyme-associated huge protein (FLASH)[26,28,29]. Similarly direct interaction of ARS2 with Drosha is thought to link CBC-ARS2 to the miRNA processing pathway[19,26]. Higher-order complexes of CBC-ARS2 involving the zinc finger protein ZC3H18 target short-lived transcripts (e.g., PROMTS) or read-through products, via the NEXT (nuclear exosome targeting) complex, for degradation by the nuclear exosome[27,30–32]. However, the structural and mechanistic details of how ARS2 fulfils these multiple functions is largely unknown, although a partial crystal structure of the *A. thaliana* homologue SERRATE exists[25] and a preliminary functional dissection of the human ARS2 domain structure has been made[26].

NELF is a multi-subunit, multi-functional complex containing subunits NELF-A, NELF-B, NELF-C (or splice variant NELF-D), and NELF-E[33,34]. Its most well-known function is, together with DSIF (DRB sensitivity-inducing factor), to induce promoter-proximal pausing of Pol II. While paused, the transcript 5′-end is capped before the switch to the processive elongation phase of transcription[35,36]. NELF-A and NELF-C have been recently shown to form the stable core of the complex with NELF-A containing a Pol II binding site and the other three subunits all having RNA binding capacity[37]. In particular, NELF-E contains an RRM domain[38], which preferentially binds a specific RNA motif (denoted NBE, NELF-E binding element) that, in

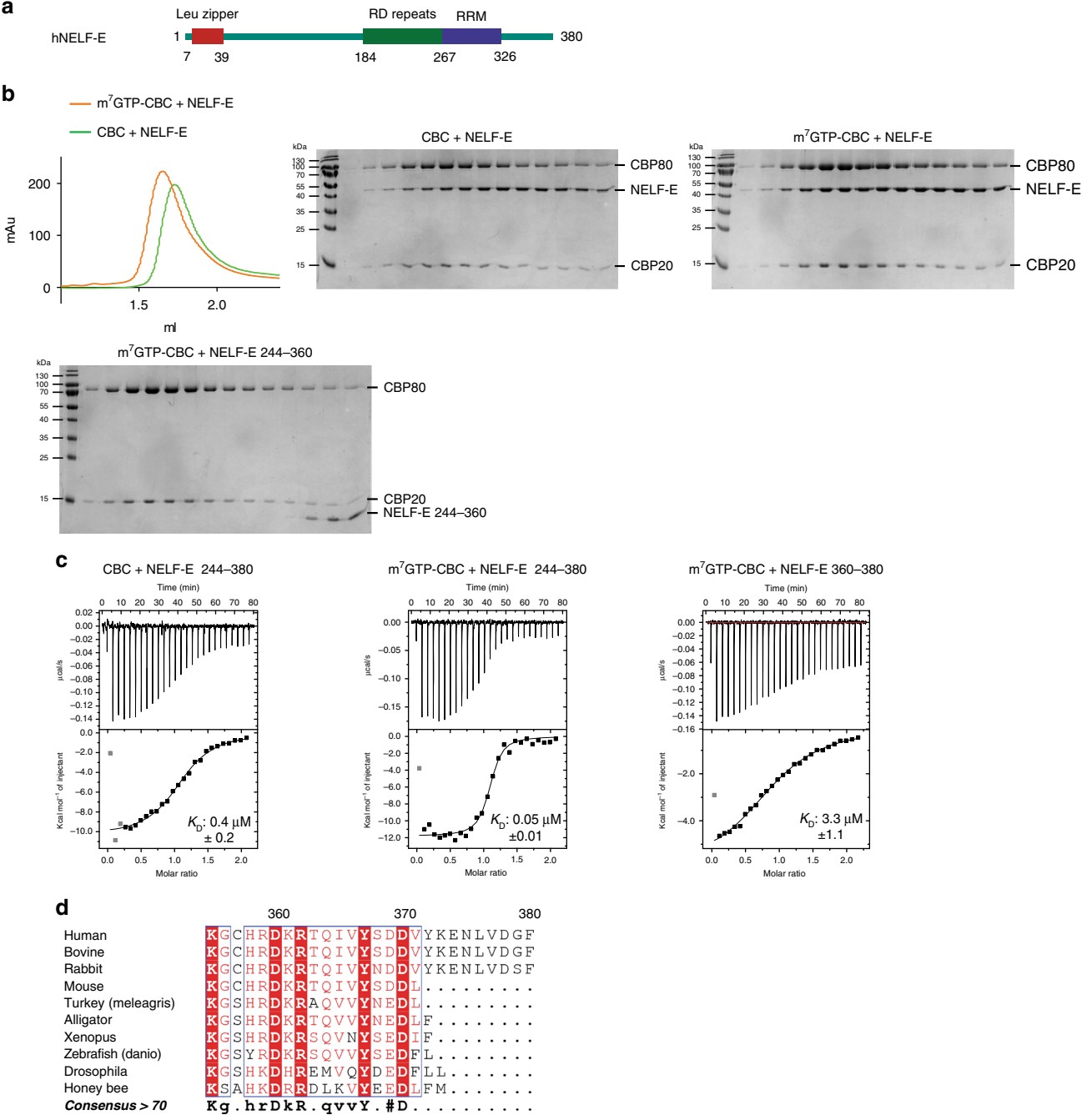

**Fig. 2** The interaction of CBC with NELF-E is enhanced in the presence of m[7]GTP. **a** Schematic diagram of the domain structure of NELF-E. **b** Gel filtration chromatogram and Coomassie-stained SDS-PAGE of CBC-NELF-E in the presence and absence of m[7]GTP. Purified recombinant CBC and NELF-E were mixed with (orange) or without (green) 500 μM m[7]GTP and subjected to gel filtration. Their elution profiles were overlaid (left) and single fractions analysed by SDS-PAGE (middle and right). Purified recombinant CBC and NELF-E[244–360] were mixed together with 500 μM m[7]GTP and similarly analysed (bottom left). **c** ITC data and curve fit to derive the affinity of different NELF-E constructs for CBC. CBC or m[7]GTP-CBC in the sample cell was titrated by NELF-E[244–380] or NELF-E[360–380]. The data were presented as described in the caption to Fig. 1c. **d** Sequence alignment of the C-terminus of NELF-E. Conserved residues are highlighted in red. Figure made with ESPript[54]

*Drosophila*, is enriched in promoter proximal regions of transcripts (+20 to +30). This suggests that NELF-E binding to an NBE may regulate pausing in a transcript-specific manner[37–39]. Furthermore, NELF-E directly binds, via its C-terminal extremity, to CBC[34]. Knockdown of either CBP80 or NELF-E in HeLa cells leads to an abnormal accumulation of aberrant poly-adenylated histone mRNAs[34]. This suggests that CBP80 and NELF-E, in conjunction with other factors such as SLBP (stem-loop binding protein), with which CBC also interacts, are important in the correct choice of cleavage site and hence termination mechanism for histone mRNAs[34]. More recently, the role of NELF in correct 3′-end processing has been extended to snRNAs with the demonstration that NELF and DSIF interact with the large, multiprotein integrator complex, which is responsible for 3′-end processing of pre-snRNAs[40–42].

The above-mentioned results show that ARS2 and NELF-E both interact directly with CBC and both are important for correct histone mRNA and snRNA processing. To understand in

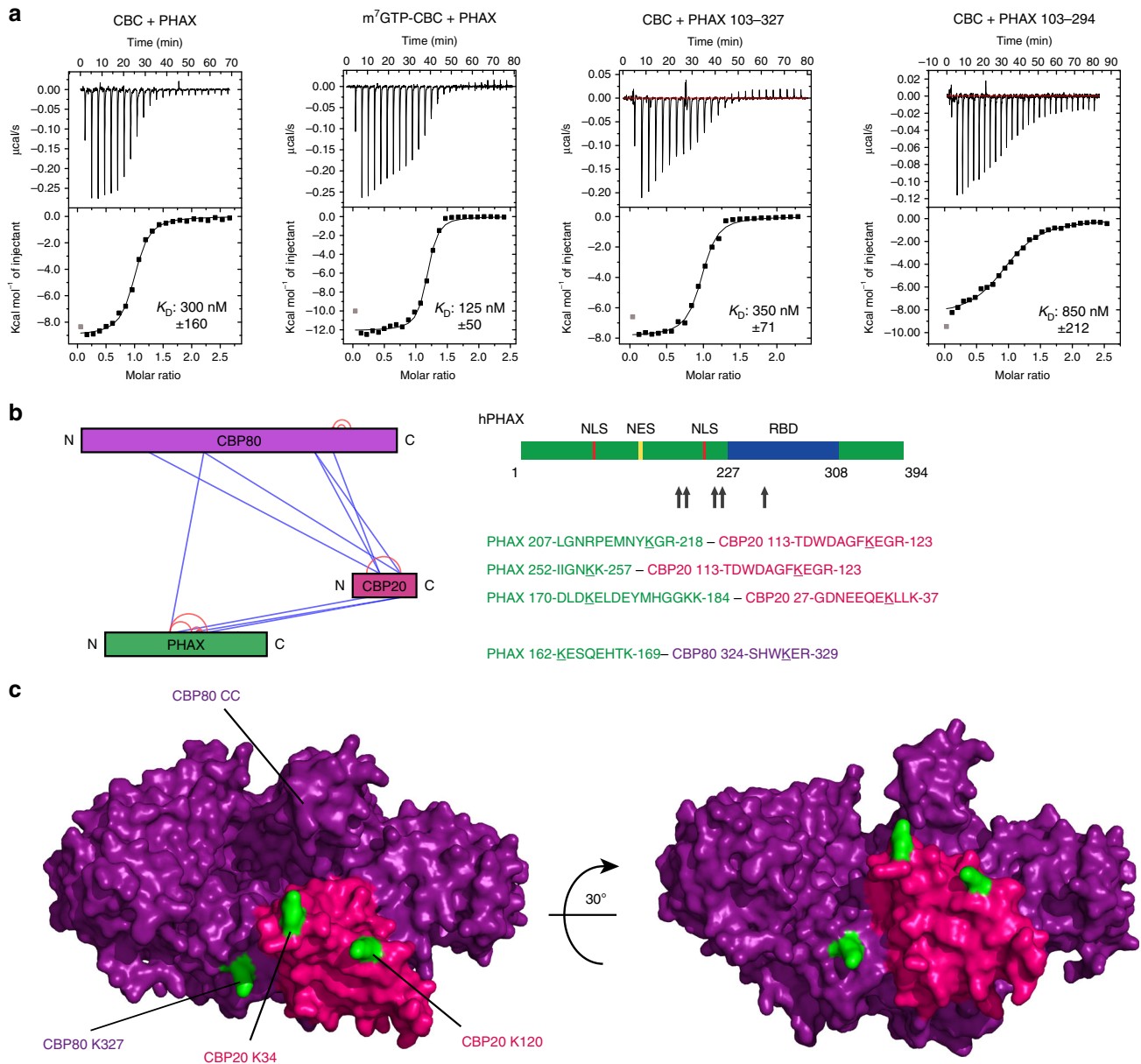

**Fig. 3** Characterisation of the PHAX–CBC interaction. **a** ITC data and curve fit to derive the affinity of PHAX constructs for CBC. CBC or m⁷GTP-CBC in the sample cell was titrated by PHAX. The data were plotted as described in the caption to Fig. 1c. **b** Identification of cross-linked lysines between CBC and PHAX by XL-MS. The CBC–PHAX complex was purified by SEC and cross-linked using DSS. After digestion and purification by SEC the cross-linked peptides were identified using mass spectrometry. **c** Location of the identified PHAX cross-linked lysines (green) on the surface of CBC (CBP80 purple, CBP20 pink, PDB: 1H2V)

more detail the molecular mechanisms involved in these processes, we set out to determine by biochemical and biophysical methods, which of ARS2, NELF-E, and PHAX can bind simultaneously to CBC. We also determine crystal structures of CBC bound to ARS2 or NELF-E peptide that explain why CBC–ARS2–PHAX and CBC–NELF–E form mutually exclusive complexes. Finally, we discuss the implications of these results in terms of remodelling of CBC complexes during co-transcriptional nuclear RNA processing.

## Results

**ARS2 C-terminus interacts with CBC in a cap-dependent manner.** We first investigated the binding of human ARS2 (isoform 4, 871 residues) to CBC in vitro. Since full-length ARS2 precipitated during purification, we used a construct denoted

ARS2$^{147–871}$, which lacks the putatively unstructured N-terminal region of the protein (Fig. 1a). Size exclusion chromatography (SEC) experiments showed that CBC and ARS2$^{147–871}$ only co-eluted in the presence of the cap analogue m⁷GTP (Fig. 1b). It has been previously shown that only residues 502–871 of ARS2 are required for interaction with CBC[24]. To further narrow down the interacting region, various truncated constructs of ARS2 were tested for their CBC binding by gel filtration and isothermal titration calorimetry (ITC). In the presence of m⁷GTP, a peptide comprising the C-terminal 27 amino acids (ARS2$^{845–871}$) was able to bind CBC, while constructs lacking these amino acids, ARS2$^{147–845}$ or ARS2$^{763–845}$ failed to bind (Fig. 1b, c). For ITC measurements we used the well behaved ARS2$^{827–871}$ construct and determined its dissociation constant ($K_D$) with respect to CBC to be $K_D$ ~12 μM in the absence and ~1 μM in the presence of the cap analogue m⁷GTP (Fig. 1c). The $K_D$ for the binding of

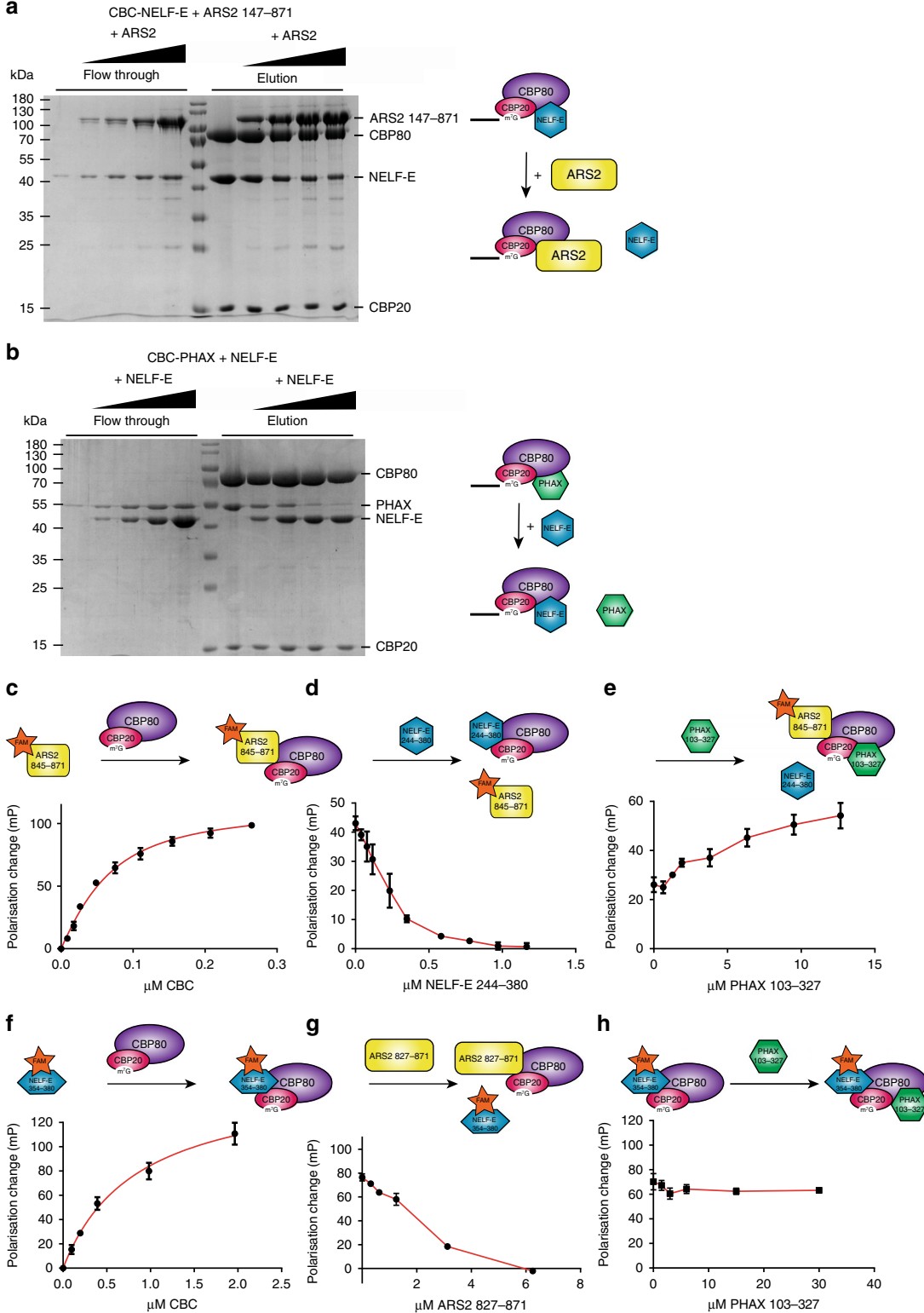

**Fig. 4** Incompatibility of ARS2-PHAX and NELF-E binding to CBC. **a**, **b** Competition on m[7]GTP-sepharose. **a** In separate experiments, different amounts of ARS2[147–871] were added to preformed CBC–NELF–E complex immobilised on m[7]GTP-sepharose. After incubation, flow through and elution were analysed by SDS-PAGE. **b** In separate experiments, different amounts of NELF-E[1–380] were added to preformed CBC–PHAX complex immobilised on m[7]GTP-sepharose. After incubation, flow through and elution were analysed by SDS-PAGE. **c–h** Competition revealed by sequential fluorescence polarisation analysis using FAM-labelled C-terminal peptides of ARS2 or NELF-E. **c** m[7]GTP-CBC titrated to ARS2[845–871](FAM). **d** NELF-E[244–380] titrated to m[7]GTP-CBC-ARS2[845–871](FAM). **e** PHAX[103–327] titrated to m[7]GTP-CBC-ARS2[845–871](FAM)/NELF-E[244–380]. **f** m[7]GTP-CBC titrated to NELF-E[354–380](FAM). **g** ARS2[827–871] titrated to m[7]GTP-CBC-NELF-E[354–380](FAM). **h** PHAX[103–327] titrated to m[7]GTP-CBC-NELF-E[354–380](FAM). The fluorophore FAM is represented with a star. Note that different constructs of NELF-E and ARS2 were used in **c**, **d**, and **e** compared to **f**, **g**, and **h** and are therefore represented by slightly different shapes. Error bars show the SD of three experiments

construct ARS2$^{845-871}$ or ARS2$^{763-871}$ to m$^7$GTP-CBC was also ~1 μM (Table 1, which gives a summary of all $K_D$'s measured in this work). These results show that the highly conserved C-terminal 27 residues (Fig. 1d) are sufficient to mediate binding of ARS2 to CBC and that m$^7$GTP binding to CBC results in a twelve-fold more stable complex in vitro.

**NELF-E C-terminus interacts with CBC cap dependently.** We next studied the in vitro interaction between CBC and NELF-E. It has previously been shown that a C-terminal NELF-E fragment (NELF-E$^{244-380}$), which includes the RRM domain of NELF-E (Fig. 2a), is able to bind CBC and that the deletion of the last C-terminal 20 amino acids abolishes the binding[34]. Purified full-length NELF-E$^{1-380}$ was subjected to SEC after incubation with CBC (Fig. 2b). In the absence of m$^7$GTP, CBC and NELF-E$^{1-380}$ partially co-eluted (Fig. 2b, middle), indicating a weak interaction of the two proteins. However, as for ARS2, addition of m$^7$GTP stabilised the complex (Fig. 2b, right). ITC experiments with the *E. coli* expressed NELF-E$^{244-380}$ construct confirmed an eight times higher affinity in the presence ($K_D$ ~0.05 μM) than in the absence of m$^7$GTP ($K_D$ ~0.4 μM) (Fig. 2c). To determine whether the C-terminal residues are sufficient for CBC binding, we used 27-mer or 21-mer synthetic peptides comprising NELF-E$^{354-380}$ or NELF-E$^{360-380}$, respectively. By ITC we determined a $K_D$ ~3.3 μM for NELF-E$^{360-380}$ binding to CBC, in the presence of m$^7$GTP (Fig. 2c). The significantly weaker interaction of NELF-E$^{360-380}$ to m$^7$GTP-CBC compared to NELF-E$^{244-380}$ ($K_D$ ~0.04 μM) suggests that the CBC–NELF–E complex is stabilised by additional interactions with CBC. However using NELF-E$^{244-360}$ that lacks the last 20 residues, no stable complex is observed by SEC, even in the presence of m$^7$GTP (Fig. 2b, lower panel). As for ARS2, the C-terminus of NELF-E is highly conserved, but interestingly, only mammals (except rodents) have an eight residue extension (Fig. 2d).

**PHAX binds to CBC independently of cap.** It has previously been shown that CBC forms a binary complex with PHAX that is stabilised in the presence of capped RNA[7,10]. Whereas the RNA binding domain (RBD) of PHAX has been well characterised[7,8], the CBC interacting region of PHAX has hitherto remained ill-defined[7]. Interestingly, much of PHAX is thought to be intrinsically disordered and even the RBD only acquires a defined tertiary structure when bound to RNA[8]. We find that the CBC–PHAX$^{1-394}$ complex can be co-purified using SEC in the absence of m$^7$GTP (Supplementary Fig. 1, left), suggesting a stronger direct interaction than for ARS2. This is confirmed by ITC which shows a $K_D$ for PHAX binding to CBC of 0.3 μM, decreasing to 0.125 μM in the presence of m$^7$GTP (eight times stronger than for m$^7$GTP–CBC–ARS2) (Fig. 3a). As assayed by SEC, the minimal CBC binding region of PHAX could be narrowed down to residues 103–294, containing most of the structured region of the PHAX RBD (227–308) (Supplementary Fig. 1, right). Using ITC it was confirmed that PHAX$^{103-327}$ had a $K_D$ of 0.35 μM (in the absence of m$^7$GTP), comparable to full-length PHAX, whereas PHAX$^{103-308}$ and PHAX$^{103-294}$ had slightly reduced $K_D$s of 0.6 and 0.85 μM, respectively (Fig. 3a; Table 1).

Since all attempts so far to co-crystallise CBC and PHAX have failed, we used chemical cross-linking coupled to mass spectrometry (XL-MS) to define lysine residues in or close to the binding sites on both proteins. The cross-links obtained (Supplementary Table 1) included some between PHAX and both subunits of CBC (Fig. 3b), consistent with previous results that both CBP20 and CBP80 are required for PHAX binding[24]. Mapped on to the CBC structure, the cross-linked positions define a region on one side of the complex where PHAX may bind (Fig. 3c).

Furthermore all PHAX lysines that are cross-linked, which span from residues 162–252, are within the minimal region of PHAX defined by truncation to be necessary for the interaction with CBC.

**NELF-E competes with ARS2 and PHAX for binding to CBC.** We next investigated which of the three CBC interacting proteins, PHAX, ARS2, and NELF-E, can bind simultaneously to CBC. It is well established that CBC, PHAX and ARS2 can form a ternary complex[24]. To test whether ARS2 and NELF-E can simultaneously bind to CBC we immobilised preformed CBC–NELF-E$^{1-380}$ complexes on m$^7$GTP-sepharose and incubated the complex with increasing amounts of ARS2$^{147-871}$. The protein bound to the resin and the flow through for each sample were then analysed by SDS-PAGE (Fig. 4a). The results show that with increasing concentration, ARS2 could displace NELF-E from the immobilised m$^7$GTP–CBC–NELF–E complex. This competition could also be observed in a fluorescence polarisation assay using FAM-labelled ARS2$^{845-871}$ or NELF-E$^{354-380}$ C-terminal peptides. As expected, addition of either of these labelled peptides to m$^7$GTP-CBC led to an increase in fluorescence polarisation, indicating binding (Fig. 4c, f). Further addition of unlabelled NELF-E$^{244-380}$ to m$^7$GTP-CBC-ARS2$^{845-871}$(FAM) or unlabelled ARS2$^{827-871}$ to m$^7$GTP-CBC-NELF-E$^{354-380}$(FAM) led to a decrease in polarisation indicating the competitive dissociation of the FAM-labelled peptide from m$^7$GTP-CBC (Fig. 4d, g).

To extend these studies to include PHAX, we used full-length PHAX$^{1-394}$ in gel filtration studies, or, in the fluorescence polarisation assay, the PHAX$^{103-327}$ construct that includes the full PHAX RBD and shows the same $K_D$ for CBC as full-length PHAX (Fig. 3a). As expected, a stable CBC–ARS2–PHAX complex can be separated using SEC, but no such complex containing CBC, NELF, and PHAX was observed (Supplementary Fig. 2). Furthermore, addition of NELF-E$^{1-380}$ to preformed CBC-PHAX$^{1-394}$ bound to m$^7$GTP-sepharose resulted in the displacement of PHAX$^{1-394}$ (Fig. 4b). Also in the presence of ARS2$^{827-871}$(FAM) PHAX$^{103-327}$ was able to displace NELF-E$^{244-380}$ from m$^7$GTP-CBC allowing CBC to rebind ARS2$^{827-871}$ (FAM), as shown by the increase of polarisation (Fig. 4e). Interestingly, titration of PHAX$^{103-327}$ to m$^7$GTP-CBC-NELF-E$^{354-380}$(FAM) did not change the polarisation (Fig. 4h). Thus, the binding of PHAX to CBC is compatible with just the NELF-E C-terminal peptide 354–380, but not with the extended NELF-E$^{244-380}$ construct, which includes the RRM domain and which binds with higher affinity to CBC (Fig. 2c). Control experiments in the absence of CBC showed that the changes in polarisation were due to interactions of the labelled peptides with CBC and not due to interactions of NELF-E or PHAX with ARS2$^{845-871}$ (FAM), or interactions of ARS2 or PHAX with NELF-E$^{354-380}$ (FAM) (Supplementary Fig. 3A, B). Also the titration of PHAX$^{103-327}$ to m$^7$GTP-CBC-ARS2$^{845-871}$(FAM) did not affect the polarisation (Supplementary Fig. 3C), confirming that the increase in polarisation when PHAX$^{103-327}$ is titrated to m$^7$GTP-CBC-NELF-E$^{244-380}$ in the presence of ARS2$^{845-871}$(FAM) is due to PHAX displacing NELF-E, thus allowing ARS2$^{845-871}$(FAM) to rebind to CBC.

**Crystal structure of m$^7$GTP–CBC–ARS2$^{827-871}$ complex.** It has been previously shown that ARS2 requires both subunits of CBC for interaction[24] and our results show that the interaction requires the C-terminus of ARS2 and is stabilised by m$^7$GTP. Therefore to obtain a structure of the CBC–ARS2 complex, we co-crystallised m$^7$GTP-CBC (CBP20-CBP80ΔNLS) with ARS2$^{827-871}$. A *P*1 crystal form diffracting to 2.8 Å resolution showed convincing ARS2 peptide density. The structure was

**Table 2 Crystallographic data collection and refinement statistics**

| Crystal | CBC ΔNLS m$^7$GTP NELF-E$^{360-380}$ | CBC ΔNLS m$^7$GTP ARS2$^{827-871}$ |
|---|---|---|
| *Diffraction data* | | |
| Beamline | ID30A1 | ID23-1 |
| Wavelength (Å) | 0.966 | 0.97917 |
| Space group | P2$_1$ | P1 |
| Cell dimensions (Å) | a = 113.8 b = 147.2 c = 153.9 α = γ = 90° β = 91.48° | a = 70.52 b = 112.99 c = 270.98 α = γ = 90° β = 90.30° |
| Resolution range of data (last shell) (Å) | 50.0–2.79 (2.87–2.79) | 50.0–2.80 (2.91–2.80) |
| Completeness (last shell) (%) | 98.4 (91.6) | 97.5 (96.3) |
| R-sym (last shell) (%) | 10.8 (129.2) | 9.7 (69.7) |
| I/σI (last shell) | 9.2 (0.91) | 6.63 (1.20) |
| CC(1/2) (last shell) | 0.995 (0.363) | 0.995 (0.585) |
| Redundancy (last shell) | 2.71 (2.70) | 2.08 (2.00) |
| *Refinement* | | |
| Reflections used in refinement work (free) | 117,089 (6345) | 190,778 (10,059) |
| R-work (last shell) | 0.202 (0.389) | 0.231 (0.386) |
| R-free (last shell) | 0.229 (0.394) | 0.268 (0.378) |
| Number of non-hydrogen atoms | 29,296 | 58,750 |
| Protein | 28,756 (4x CBC) | 57,548 (8x CBC) |
| Ligand | 132 (4x m$^7$GTP) | 264 (8x m$^7$GTP) |
| Peptide | 408 (chains E, K, Z) | 938 (chains C, F, I, L, O, R, U, X) |
| *Geometry and B-factors* | | |
| RMSD (bonds) (Å) | 0.009 | 0.008 |
| RMSD (angles) (°) | 1.312 | 1.194 |
| Ramachandran favoured (%) | 97.0 | 97.9 |
| Ramachandran outliers (%) | 0.2 | 0.03 |
| Clash score | 1.03 | 0.86 |
| MolProbity score | 1.17 | 0.88 |
| Average B-factor | 74.4 | 73.3 |
| Protein | 74.2 (4× CBC) | 72.8 (8×CBC) |
| Ligand | 70.7 (4× m$^7$GTP) | 66.3 (8× m$^7$GTP) |
| Peptide | 111.5 (chains E, K, Z) | 106.1 (chains C, F, I, L, O, R, U, X) |

solved by molecular replacement using the known m$^7$GpppG–CBC complex structures (PDB:1H2T) as search model. There are eight complexes in the asymmetric unit with variable occupancy of the ARS2 peptide, probably due to the relatively low affinity. Data collection and refinement statistics are given in Table 2. Electron density for one example of the peptide is shown in Supplementary Fig. 4A. The overall structure of CBC is only perturbed locally by ARS2 binding (see below) and the binding of the cap analogue m$^7$GTP to CBP20 is unchanged from described previously[2].

The C-terminus of ARS2, which is only visible beyond residue 851, forms an extended chain that wraps around the CBP80–CBP20 interface, interacting with both subunits (Fig. 5a). The total buried surface area upon ARS2 peptide binding to CBC is 2257 Å$^2$ (PISA server, www.ebi.ac.uk/pdbe/pisa/) shared 52 and 48%, respectively, between CBP80 and CBP20. The principle interacting regions of CBP80 are 460–463, 557–570, 607–616, and 647–651, and of CBP20 are 48–55 (β1–α2 loop of the RRM domain) and 71–80 (β2–β3 loop). There are two main interacting regions of the ARS2 peptide, residues 852-DPRAIVEYRD-861, notably Asp852, Arg854, and Tyr859, which bind in the

"proximal" site and the extreme C-terminal residues 868-VDFF-871, notably Phe871, which bind in the "distal" site (Fig. 5a, b). The electron density is absent for the intervening residues 862-LDAPDD-867. In the proximal site, Asp852 electrostatically stabilises Arg854, whose guanidinium group stacks on Tyr461 of CBP80 and forms a salt bridge with Asp107 from CBP20 (Fig. 5b, left). In the absence of the ARS2 peptide, the hydroxyls of CBP80 Tyr461 and CBP20 Tyr50 interact at the inter-subunit interface[2]. However, this is sterically incompatible with the presence of the side chains Asp852 and Arg854 of ARS2 with the result that CBP20 Tyr50 is stabilised in an alternative rotamer that allows it to stack with ARS2 Ile856 (Fig. 5b, left). Tyr859 is inserted between the subunits making contacts with CBP20 Met71 and Leu73 and CBP80 Ser558 and His561 (Fig. 5b, middle). Its hydroxyl points toward the main-chain amino group of CBP20 Glu53, whose side chain makes CBC inter-subunit interactions. To assess potential global changes in CBC structure we superposed, by means of CBP80, the m$^7$GpppG–CBC complex (PDB:1H2T or 1H2U) on the complex with ARS2 (RMSD 0.35 Å for 1H2T and 0.38 Å for 1H2U). The superposition shows that the CBP20 β2–β3 loop (residues 71–81) and C-terminal extension to the RRM domain (residues 126–150, that are only ordered upon m$^7$GTP binding) are slightly shifted toward CBP80, closing the groove in which the ARS2 peptide binds (maximum backbone displacement of 2.0 Å at CBP20 Lys77) (Supplementary Fig. 5). This allows the main-chain of CBP20 residues 78–80 to run anti-parallel to the ARS2 residues 857–859 with putative hydrogen bonds between the carbonyl oxygen of Lys78 with the amide of Tyr859 and that of Val857 with the amide of Ala80. In the distal site, the side chain of Phe871, the extreme C-terminal ARS2 residue, is buried in a hydrophobic pocket formed by CBP80 Ile609, Cys616, Met648, and His651 at the base of the long, solvent protruding coiled-coil of CBP80 (residues 650–705) (Fig. 5b, right). The side chain of His651 partially stacks on Phe871 and also interacts electro-statically with the C-terminal carboxylate of ARS2 (Fig. 5b, right). This binding region is also at the inter-subunit interface, with CBP20 residue Tyr89 hydrogen bonding to ARS2 Asp869 as well as contacting CBP80 Arg610, whose carbonyl-oxygen hydrogen bonds to the amides of ARS2 Phe870 and Phe871 (Fig. 5b, right). All of the cited interacting residues are highly conserved in metazoan ARS2 (Fig. 2d) or CBC, suggesting conservation of this interaction and underscoring its biological importance.

To validate the observed mode of interaction, point mutations in ARS2 and both subunits of CBC were introduced by site directed mutagenesis and tested for CBC–ARS2 complex formation (Fig. 5c). Mutations were made in the *E. coli* expressed ARS2$^{827-871}$ construct in three of the key interacting ARS2 residues, Arg854, Tyr859, and Phe871. ITC measurements showed that the R854A/Y859A double mutation severely weakened the interaction with CBC, whereas F871D alone did not have a major effect (Fig. 5c). For CBC, a triple mutant in CBP80 (Y461A/R610E/H651A) was combined with various mutations in CBP20 (Y50A, Y50A/Y89A, or Y50A/Y89A/ D107R). Binding of ARS2$^{827-871}$ to mutated CBC was first tested by m$^7$GTP-sepharose pull down (Supplementary Fig. 6). The combination of the triple mutants of both CBP20 and CBP80 abolished co-elution with ARS2$^{827-871}$, but this mutated CBC showed a weaker binding to the m$^7$GTP resin. This was confirmed by ITC experiments which gave for the m$^7$GTP±CBC interaction a $K_D$ of 70 μM for this mutant compared to a $K_D$ of 0.1 μM for wild-type CBC (Table 1). On the other hand, CBC with the single Y50A mutation in CBP20 together with the CBP80 triple mutation Y461A/R610E/H651A (denoted CBCmut) bound m$^7$GTP and PHAX as well as wild-type CBC, confirming that CBC containing CBCmut was still functional. This was also

the case when CBP20 had the double mutation Y50A/Y89A. ITC showed a twelvefold reduction in affinity of CBCmut for ARS2$^{827-871}$ ($K_D$ ~12 μM) compared to wild-type CBC ($K_D$ ~1 μM) (Fig. 5c, right).

**NELF-E and ARS2 have the same mode of binding to m$^7$GTP-CBC**. Similarly to the CBC–ARS2 complex, crystals of CBC-NELF-E$^{360-380}$ or CBC-NELF-E$^{354-380}$ could be obtained in the presence of m$^7$GTP or m$^7$GpppG. They showed varying space

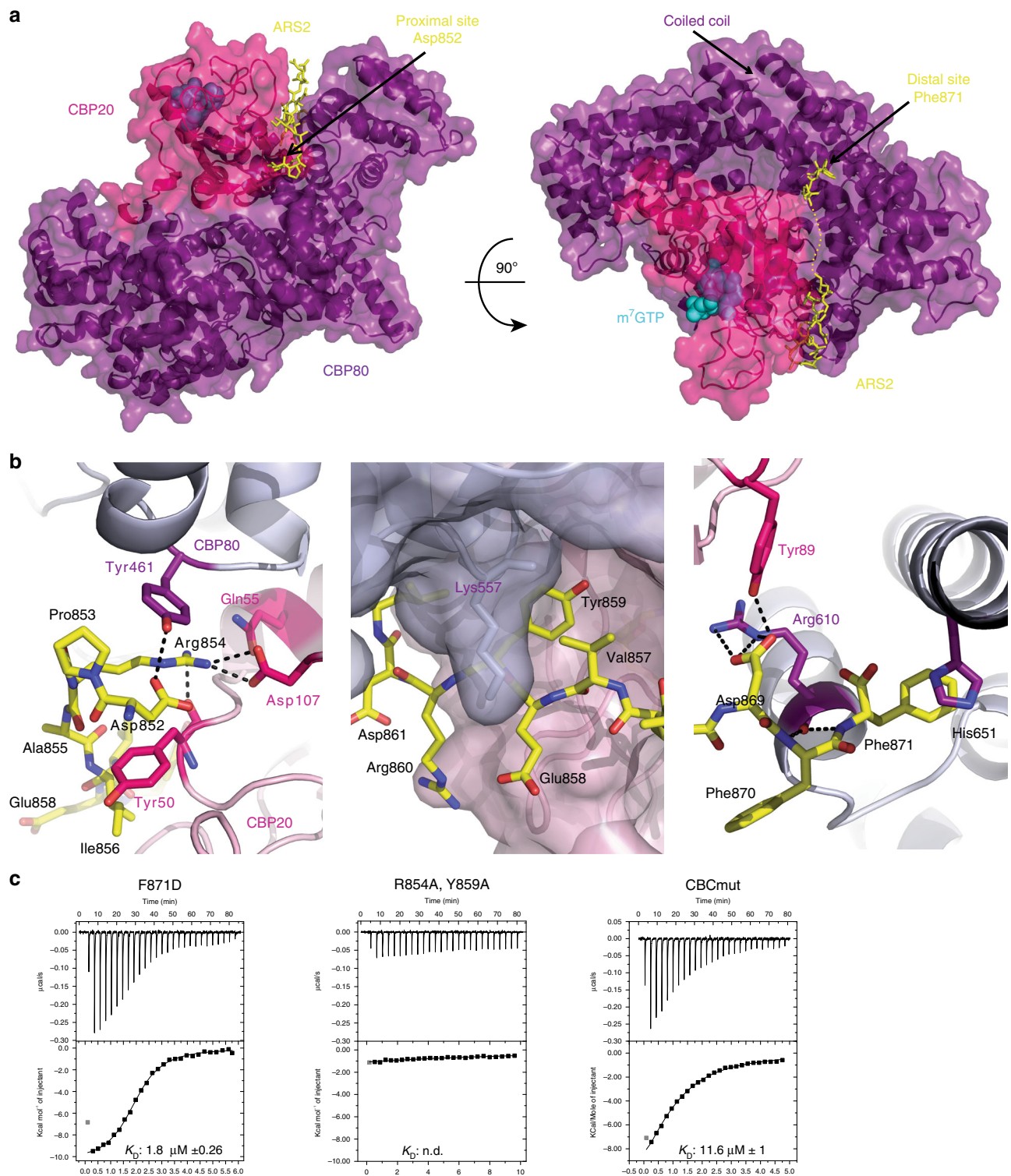

**Fig. 5** Crystal structure of the CBC–ARS2$^{827-871}$ complex. **a** Crystal structure of m$^7$GTP-CBC-ARS2$^{827-871}$. CBC is depicted as a semi-transparent surface with underlying secondary structure as ribbons with CBP20 and CBP80 in pink and purple, respectively. ARS2$^{852-871}$ is shown as yellow sticks and m$^7$GTP as cyan spheres. **b** Zoomed view of the CBC–ARS2 interface highlighting key ARS2 interacting residues Arg854 (left), Tyr859 (middle), and Phe871 (right). Putative hydrogen bonds are indicated with black dotted lines. **c** ITC data of ARS2$^{827-871}$ mutants and CBC and ARS2$^{827-871}$ and CBCmut in the presence of m$^7$GTP. m$^7$GTP-CBC in the sample cell was titrated by ARS2$^{827-871}$. The data were presented as described in the caption to Fig. 1c. $K_D$ values represent the average from at least two independent experiments (Table 1)

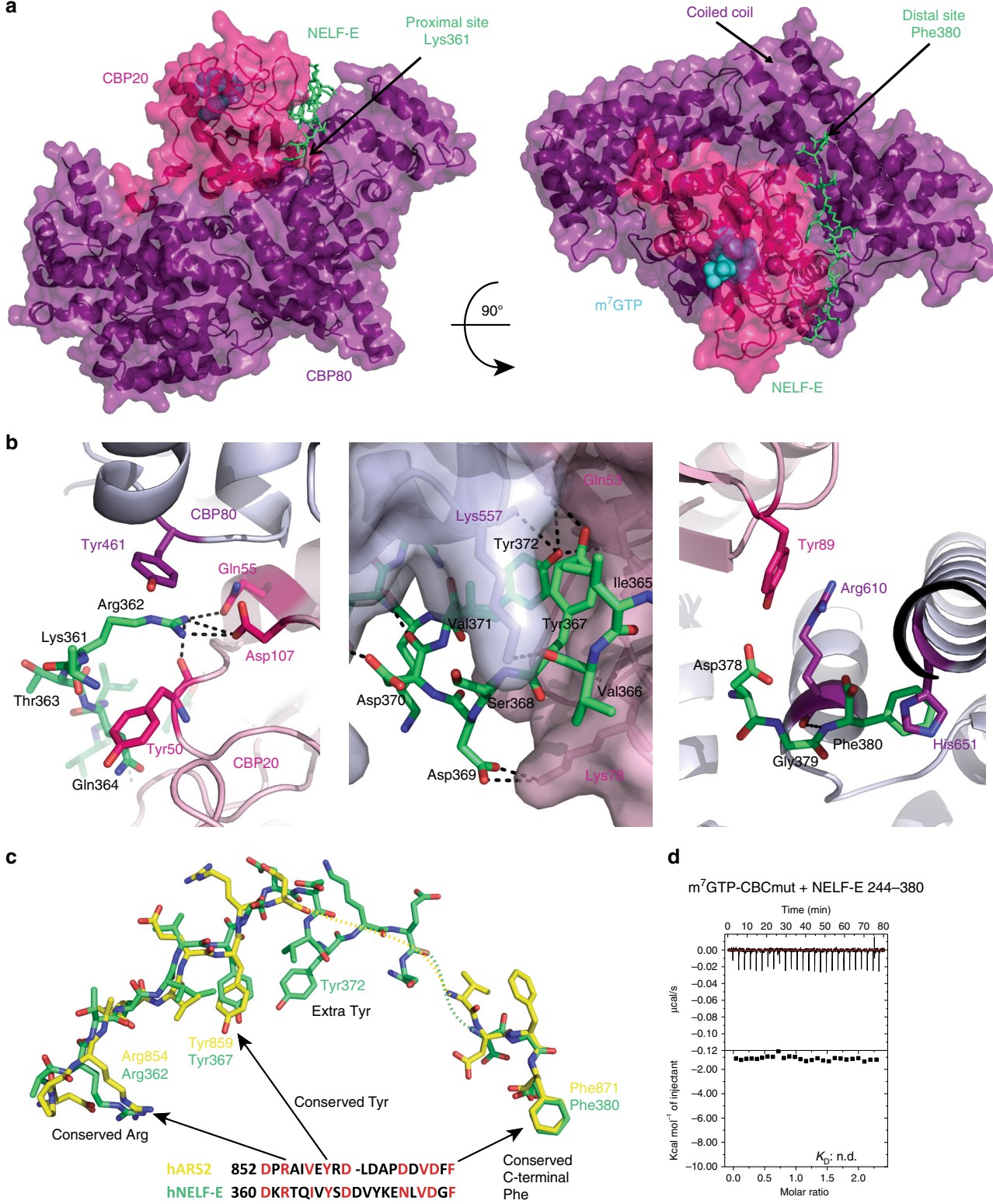

**Fig. 6** Crystal structure of the CBC–NELF-E$^{360-380}$ complex. **a** Crystal structure of m$^7$GTP-CBC-NELF-E$^{360-380}$. CBC is depicted as a semi-transparent surface with underlying secondary structure as ribbons with CBP20 and CBP80 in pink and purple, respectively. NELF-E$^{361-380}$ is shown as green sticks and m$^7$GTP as cyan spheres. **b** Zoom-in view of residues at the CBC–NELF-E interface highlighting key NELF-E interacting residues Arg362 (left), Tyr367 and Tyr372 (middle), and Phe380 (right). Putative hydrogen bonds are indicated with black dotted lines. **c** Comparison of the conformation of NELF-E (green) and ARS2 (yellow) C-terminal peptides after superposition of CBP80 of both complexes. A sequence alignment of the two peptides is also shown, with the conserved residues in red. **d** ITC data of m$^7$GTP-CBCmut and NELF-E$^{244-380}$. m$^7$GTP-CBCmut in the sample cell was titrated by NELF-E$^{244-380}$. The data were presented as described in the caption to Fig. 1c. $K_D$ values represent the average from at least two independent experiments (Table 1)

groups but generally very weak density for the NELF-E peptide. The best results were obtained with CBC-NELF-E$^{360-380}$ and m$^7$GTP giving $P2_1$ monoclinic crystals that diffracted to 2.8 Å resolution (Table 2). There are four m$^7$GTP bound CBC complexes in the asymmetric unit with clear electron density for the NELF-E$^{360-380}$ peptide in one of them (chain E, Supplementary Fig. 4B) but weaker in the other copies.

Remarkably the NELF-E and ARS2 C-terminal peptides bind to exactly the same site on m$^7$GTP-CBC, at the interface of CBP20 and CBP80 (Fig. 6a) and induce the same slight displacement of interacting regions of CBP20. The total buried surface area upon the NELF-E peptide binding to CBC is 2501 Å$^2$ (PISA server), again shared equally between each CBC subunit. The NELF-E peptide has a similar bipartite binding site to that of ARS2, with 360-DK**R**TQIV**Y**SDDVYKE-372 in the proximal site and 378-DG**F**-380 in the distal site, and the bridging residues 375-NLV-377 poorly ordered. NELF-E Arg362 makes nearly identical interactions with CBP80 Tyr461 and CBP20 Gln55 and Asp107 as does Arg854 of ARS2 (Fig. 6b, left). NELF-E Tyr367 (which partially stacks on NELF-E Ile365) is buried in the same pocket as ARS2 Tyr859 (which partially stacks on ARS2 Val857), with its hydroxyl making an interaction with the side chain of CBP20 Glu53 (Fig. 6b, middle). Finally, there are highly analogous interactions of the NELF-E C-terminal Phe380 with CBP80 His651 as observed for ARS2 Phe871 (Fig. 6b, right). Sequence alignment of the NELF-E and ARS2 C-terminal peptides highlights a remarkable sequence homology that was not recognised before this structural analysis, with 7 identical residues out of 21, including those that in both cases make the key interactions with CBC (Fig. 6c). The major difference between NELF-E and ARS2 binding is that NELF-E has an additional tyrosine (Tyr372) which is buried in the interface. This tyrosine makes van der Waals contacts on one side with NELF-E Val371, and on the other side with CBP20 Ile70 and its hydroxyl group hydrogen bonds to the main-chain amino group of CBP20 Met71. As additional confirmation that NELF-E and ARS2 C-terminal peptides bind in the same groove between CBP20 and CBP80, we found by ITC that CBCmut, which significantly reduced binding to ARS2 (Fig. 5c, right), completely abolished binding to NELF-E$^{244-380}$ (Fig. 6d).

## Discussion

In this work we have used biochemical, biophysical, and structural methods to analyse the interactions between CBC and three of its direct partners, ARS2, PHAX, and NELF-E. We show that both ARS2 and NELF-E interact with CBC via their C-terminal extremities (ARS2$^{845-871}$ and NELF-E$^{360-380}$, respectively) and that this interaction is enhanced 12-fold and 8-fold, respectively, when the cap analogue m$^7$GTP is bound to CBC. In the case of ARS2 this appears to be the only interaction with CBC, whereas NELF-E$^{244-380}$ (which includes the NELF-E RRM domain) binds more strongly to CBC than just the C-terminal peptide. Binary complex formation between PHAX and CBC does not depend significantly on m$^7$GTP binding and can be substantially recapitulated with the truncated PHAX$^{103-294}$ fragment.

Our biochemical results demonstrate the existence of two mutually exclusive CBC containing complexes, CBC–ARS2–PHAX and CBC–NELF-E. Conversely they show that the putative complexes CBC–PHAX–NELF–E or CBC–ARS2–NELF–E cannot exist. Structural analysis shows that the basis for the mutually exclusive binding of ARS2 and NELF-E to CBC is that the C-terminal peptides of ARS2 and NELF-E occupy not only exactly the same binding site at the interface between CBP20 and CBP80, but also interact in the same bipartite manner. Indeed, the two peptides are homologous with three key

interacting residues being identical, Arg362, Tyr367, and Phe380 for NELF-E and Arg854, Tyr859, and Phe871 for ARS2 (Fig. 6c). The arginine and tyrosine are absolutely conserved in ARS2 and NELF-E of all metazoans and for plant ARS2 as well (plants lack NELF). The terminal phenylalanine, which binds in the distal pocket at the base of the long coiled-coil of CBC, is well conserved in metazoan ARS2 but not in plant ARS2 which have a six residue C-terminal extension (Fig. 1d). For NELF-E, most mammals, except rodents, have an eight residue C-terminal extension ending in terminal Phe380 (Fig. 2d). On the other hand, NELF-E has an additional important CBC interacting residue, Tyr372, which is conserved as a phenylalanine or leucine in most metazoan NELF-E (Fig. 2d). Consistent with these observations, we have shown for ARS2, that binding of the absolutely conserved arginine and tyrosine in the proximal pocket contributes more to the strength of interaction than the phenylalanine binding in the distal pocket (Fig. 5c). Three other observations are interesting here. First, we show that m$^7$GTP (and by extension, the 5′-cap of capped transcripts) enhances the binding of both ARS2 and NELF-E C-terminal peptides to CBC. That this synergistic binding is reciprocal was confirmed by ITC measurements showing that titrating m$^7$GTP to CBC in the presence of saturating NELF-E or ARS2 increased the affinity for cap by respectively factors of ~3 and ~7 compared to m$^7$GTP binding to CBC alone (Supplementary Fig. 7). Without structures of the low-affinity CBC–peptide complexes, we can only speculate about the precise mechanism of synergistic binding. It is well established that cap-binding stabilises the fold of the otherwise flexible N-terminal (1–32) and C-terminal (125–150) regions of CBP20[2]. Nevertheless, most of the CBP20 residues that interact with ARS2 or NELF-E are actually already structured in the absence of cap. However, superposition by means of CBP80 of the apo (PDB:1H2V), cap-bound (PDB:1H2T), and the ARS2 (or NELF-E)–CBC–m$^7$GTP complexes shows that, due to the slight relative rotation of CBP20 upon ligand binding, the peptide binding groove is closer to optimally configured when the cap is bound rather than unbound (Supplementary Fig. 5). Second, the most highly conserved motif in the ARS2 C-terminal peptide is 861-DLDAP (Fig. 1d), yet these residues lack electron density and appear not to interact with CBC. This suggests that this motif could have another functional role perhaps in a context when ARS2 is not bound to CBC. Third, the phosphosite database[43] contains strong evidence that ARS2 Tyr859 (Tyr864 in human ARS2 isoform 1) can be phosphorylated (http://www.phosphosite.org/siteAction.action?id=11169746). There is also evidence that the equivalent tyrosine in NELF-E, Tyr367, and possible also Tyr372 can be phosphorylated (http://www.phosphosite.org/proteinAction.action?id=5808). The structures suggest that these post-translational modifications would likely impede ARS2 or NELF-E binding to CBC, since the tyrosine side chains are buried, suggesting that complex formation could be regulated by phosphorylation.

Our combined binding studies suggest that the competitive binding of PHAX or NELF-E to CBC results from steric exclusion of PHAX binding by parts of NELF-E preceding its C-terminal peptide. The structural basis for this competition remains to be established since the mode of binding of PHAX or full-length NELF-E to CBC are currently unknown. The only structural information available to date on PHAX-CBC binding comes from mapping cross-linked lysines on to the CBC structure. This highlights a region involving both CBP20 and CBP80 where PHAX may bind, which is in close proximity to the ARS2/NELF-E binding site on CBC (Supplementary Fig. 8).

Although we have only focussed so far on direct protein–protein interactions between CBC and ARS2, NELF-E or PHAX, in fact all three proteins contain established (PHAX,

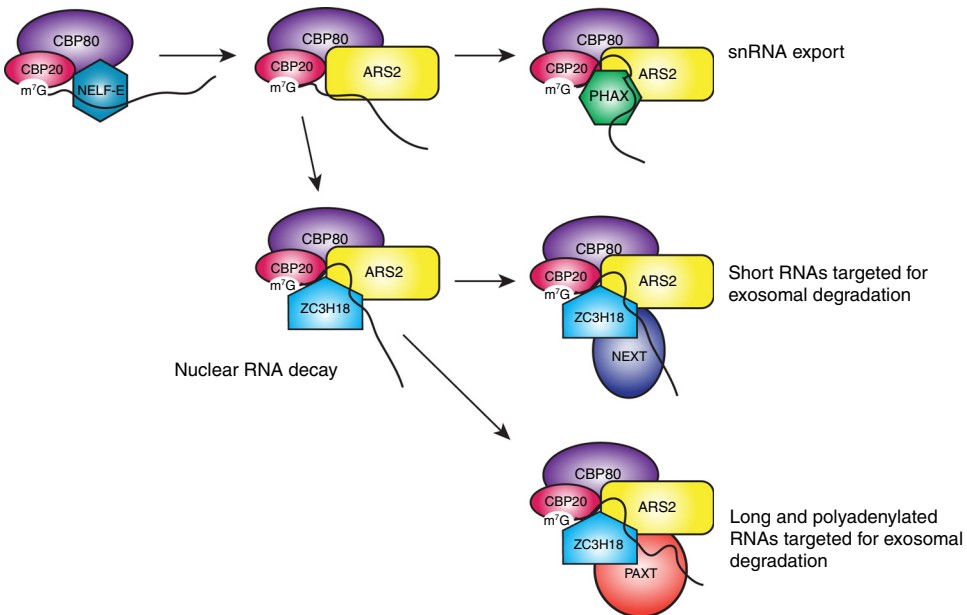

**Fig. 7** Model of mutually exclusive CBC complexes. Schematic diagram showing mutually exclusive CBC containing transcript complexes involved in transcription pausing (CBC-NELF-E), export of mature transcripts (CBC-ARS2-PHAX), or targeting for degradation of aberrant transcripts (CBC-ARS2-NEXT or CBC-ARS2-PAXT) defined by this work and that of others[24,30,44]

NELF-E) or putative (ARS2) RBDs that are presumed to interact with 5′-proximal regions of the CBC-bound-capped RNA. The capped transcript is thus an additional tether between CBC and its partner protein and this is likely to enhance the apparent interaction between the two proteins. Although quantification of this effect is beyond the scope of this work, it has been reported that snRNA binding enhances the stability of the CBC–PHAX complex[7]. Whereas PHAX RNA binding is non-sequence specific[7,8], NELF-E RRM preferentially binds a particular RNA motif[37,39]. However, the stability of CBC–NELF–E–RNA complexes remains to be studied. Similarly the RNA specificity of ARS2 currently remains ill-defined.

What can be the biological significance of the mutual exclusivity of the CBC–ARS2–PHAX and CBC–NELF–E complexes? They could be components of parallel but different pathways (i.e., acting on distinct RNAs) or be remodelled complexes at different time points in the biogenesis pathway of the same RNA. The latter certainly seems plausible since NELF-E is generally thought to be associated with Pol II pausing early after transcription initiation, whereas ARS2 and PHAX are linked to later events such as 3′-end processing[20,24,27], nuclear export[10], or nuclear exosome-mediated degradation[27,30,44]. Concerning the early co-transcriptional events, capping is thought to occur when Pol II is paused due to the action of two NELFs, DSIF and NELF[36]. In the absence of the cap, CBC still has a weak affinity to NELF-E, and as we have shown this binding enhances the affinity of CBC for cap. Thus, NELF-E could be one factor in helping to recruit CBC to the newly capped transcript. High-affinity cap-binding to CBC will subsequently strengthen the CBC–NELF–E interaction. Furthermore, the ternary CBC-NELF-E-capped RNA complex can be further reinforced by NELF-E interactions, via its RRM, with the transcript and this has been proposed to be one mechanism for regulating pausing in a transcript-specific manner[39]. Transcription pause release is promoted by multiple phosphorylation events mediated by P-TEFb[45] (which has been reported to bind to CBC[46]), including phosphorylation of DSIF (whereupon it becomes a positive elongation factor), NELF (leading to its dissociation), and the Pol II C-terminal domain. However, recent work has implicated the integrator complex at

this early stage, showing that it is recruited by NELF and DSIF to the paused Pol II complex and integrator subunits also regulate pausing[40–42]. To be more specific, we now restrict discussion to how our observations might fit into a recent model of co-transcriptional processing of snRNAs[40,44]. Bearing in mind that these are very short transcripts, these authors and others observe that NELF accumulates at the downstream regions of snRNA (and replicative histone mRNAs) genes and is required for correct 3′-end processing, likely by recruiting the integrator. They suggest that NELF has a role in preventing aberrant polyadenylation by inhibiting recruitment of the cleavage stimulation factor. Assuming NELF is still bound to CBC at this stage, the CBC–NELF–E interaction will also prevent premature recruitment of ARS2 and PHAX. Nevertheless, at some currently unknown 3′-end processing step, there must be a switch of CBC partner from NELF-E to ARS2. This may be concomitant with NELF dissociation and/or additional phosphorylation events (e.g., as mentioned above on Y367/Y372 of NELF-E). Once the switch has happened, PHAX can be recruited to the mature snRNA–CBC–ARS2 complex to promote its nuclear export. Alternatively, if there is some defect in 3′-end processing such as read-through, leading to extended forms of snRNAs, the aberrant snRNA bound to CBC-ARS2 can associate with zinc-finger protein ZC3H18 and the RBM7 subunit of the NEXT complex and be targeted for degradation[27,31,44]. Interestingly, it has recently been shown that the NEXT complex, via the bridging protein ZC3H18, interacts with CBC-ARS2 in such a way as to exclude PHAX binding[44]. Furthermore, ZC3H18 containing CBC complexes can link to either of two distinct nuclear degradation complexes, NEXT or PAXT (poly(A)-tail exosome targeting), depending on whether ZCCHC8 or ZFC3H1, respectively, are part of the complex[27,30]. Thus dynamic competition between mutually exclusive CBC complexes contributes to the productive export of mature or degradation of incorrectly processed snRNA transcripts (summarised in Fig. 7) and appears to be a pervasive mechanism in co-transcriptional processing. Further work is needed to validate this model, to define which complexes are associated with different classes of Pol II transcripts and to clarify the mechanisms that trigger complex remodelling.

## Methods

**Protein purification**. hARS2 isoform-4 (871 residues, UniProt:Q9BXP5-4, also known as isoform-e, NP_001122326) constructs (residues 147–871, 147–845, 763–871, 763–845, 827–871) hNELF-E (NM_002904.5) (244–380, 244–360) and hPHAX (NM_032177.3) (1–394, 103–294, 103–308, 103–327) were cloned into pETM11 (EMBL). Full-length hNELF-E (1–380) was cloned into pFASTBac and CBP20 into pETM30. Primers used are given in Supplementary Table 2. All constructs were expressed in *E. coli* Rosetta 2 (Novagen), except full-length NELF-E and CBP80ΔNLS (CBP80 with residues 1–19 deleted[2]), which were expressed in High Five insect cells (Life Technology). Cells were harvested and lysed by sonication in lysis buffer containing 50 mM HEPES, pH 7.8, 300 mM NaCl, 10% (v/v) glycerol, 5 mM β-mercaptoethanol. Lysates were clarified by centrifugation and applied on an equilibrated Ni-sepharose column (GE Healthcare), washed with 20 mM imidazole in lysis buffer, and eluted with 300 mM imidazole in lysis buffer. While dialysing into 20 mM HEPES, pH 7.8, 120 mM NaCl, 10% (v/v) glycerol, 5 mM β-mercaptoethanol, the N-terminal 6-histidine-tag was cleaved with His-tagged Tobacco Etch Virus (TEV) protease overnight in a ratio 1:100 mg TEV per mg protein. The cleaved protein was further purified first on Ni-sepharose to remove TEV and then using an anion exchange column (HiTrap Q, GE Healthcare) followed if necessary by a Heparin column (GE Healthcare). In both cases a NaCl gradient from 50 to 800 mM in 20 mM HEPES 7.8, 0.5 mM tris(2-carboxyethyl)phosphine was used. To reconstitute the CBC heterodimer, cleaved His-tagged CBP20 lysate was first immobilised on a Ni-sepharose column. After washing the column with lysis buffer, the cleaved CBP80 lysate was applied to the column and further purification steps performed as described above. To remove the access of CBP20 the flow through of the Ni-column after TEV cleavage was loaded on an anion exchange column (HiTrap Q, GE Healthcare) and CBC eluted by a NaCl gradient as described above. In the text we refer to CBP80ΔNLS and CBP80ΔNLS-CBP20 as CBP80 and CBC, respectively. ARS2[845–871], NELF-E[354–380], and NELF-E[360–380] peptides, with or without FAM label at the N-terminus, were purchased from GenScript.

**Isothermal titration calorimetry**. ITC experiments were performed on a MicroCal ITC200 system (Malvern). Proteins were dialyzed overnight in buffer 20 mM HEPES, 120 mM NaCl, 2 mM tris(2-carboxyethyl)phosphine, pH 7.8. Injections of 1.5 μl protein A were added at an interval of 120–180 s into 205 μl of protein B in the cell, with a stirring speed of 800 rpm at 25 °C. The experimental data were analysed with MicroCal ITC Origin and were fitted to a single-site binding equation.

**Fluorescence polarisation measurements**. Fluorescence anisotropy binding assays were performed with N-terminal FAM-labelled ARS2[845–871] and NELF-E[353–380] peptides. 40 nM peptides were titrated with increasing concentrations of proteins in 20 mM HEPES, pH 7.8, 120 mM NaCl, 2 mM tris(2-carboxyethyl) phosphine. For the competition assay CBC–ARS2 and CBC–NELF-E complexes were preformed in the presence of 25 μM m7GTP using 40 nM ARS2[827–871](FAM) peptide together with 60 nM CBC and 40 nM NELF-E[354–380](FAM) peptide with 1 μM CBC. These preformed complexes were titrated with NELF-E[244–380], ARS2[827–871], and PHAX[103–327], respectively. For the displacement assay between PHAX and NELF-E, PHAX[103–327] was titrated to a mixture of 40 nM ARS2[827–871] (FAM) peptide, 60 nM CBC, 270 nM NELF-E[244–380], and 25 μM m7GTP. To observe a change in polarisation at reasonable protein concentrations, CBC and NELF-E[244–380] concentrations that saturate half of the FAM-labelled peptides were used to preform the individual complexes. Fluorescence polarisation was measured at 25 °C with a microplate reader (CLARIOstar BMG LABTECH) using an excitation wavelength of 495 nm and emission wavelength of 515 nm. For analysis the polarisation value for the peptide alone was subtracted from the measurements.

**Displacement pull down**. CBC-NELF-E[1–380] or CBC-PHAX[1–394] were immobilised on m7GTP-sepharose and incubated for 1 h at 4 °C with different amounts of ARS2[147–871] or NELF-E[1–380] in 20 mM HEPES, pH 7.8, 120 mM NaCl, 2 mM tris(2-carboxyethyl)phosphine, respectively. The flow through, as well as the elution from the beads after extensive washing, was analysed by SDS-PAGE followed by Coomassie staining.

**Cross-link mass spectrometry**. Purified CBC–PHAX complexes were cross-linked with H12/D12 disuccinimidyl suberate (DSS, Creative Molecules Inc.) at 35 °C for 30 min. The reaction was quenched, digested with LysC (Wako) and trypsin followed by peptide separation using SEC as described previously[47]. MS data acquisition and analysis using the xQuest/xProphet software was performed according to ref. [47]. Cross-linked peptides were filtered with a delta score <0.95, a false discovery rate <0.05, and a linear discriminant (ld) confidence score of >28.

**Crystallisation**. For the CBC–ARS2 complex, numerous crystallisation trials were performed with CBC, m7GTP, or m7GpppG and ARS2 constructs/peptides 147–871, 763–871, 787–871, 811–871, 827–875, 845–871. The crystallised CBC–ARS2[827–871] complex was prepared by mixing CBC with an access of ARS2[827–871] in the presence of 1 mM m7GTP and subjecting the mixture to gel filtration (120 mM NaCl, 5 mM β-mercaptoethanol, 20 mM HEPES, pH 7.8). The complex was concentrated to 8 mg ml⁻¹ using Amicon ultra centrifugal filters (Merck Millipore). The best crystals were

obtained at 20 °C in 2 μl hanging drops with a 1:1 ratio of protein solution to crystallisation solution. The crystallisation solution contained 0.1 M sodium acetate pH 5, 8% (v/v) MPD, and 0.1 M guanidine hydrochloride.

For the CBC–NELF-E complex, numerous crystallisation trials were performed with CBC or CBCΔCC[2], m7GTP, or m7GpppG (added to CBC before setting up the drops) and CBC-NELF-E[360–380] or CBC-NELF-E[354–380]. Several different crystal forms were obtained but generally with weak density for the peptide. The best crystals were of m7GTP-CBC-NELF-E[360–380] obtained by mixing 6 mg ml⁻¹ CBC, 500 μM m7GTP, and 500 μM peptide in 120 mM NaCl, 1 mM tris(2-carboxyethyl)phosphine, 20 mM HEPES, pH 7.8. Crystals grew at 20 °C in a sitting drop set-up with 0.2 μl drops containing 1:1 ratio of mother liquor (0.2 M lithium citrate and 20% (w/v) PEG 3350) and protein.

**Structure determination**. For data collection, crystals were flash-frozen in well solution supplemented with 20% (v/v) glycerol and diffraction measured on European Synchrotron Radiation Facility (ESRF) beamlines, MASSIF-1 (ID30a-1)[48] for CBC-NELF-E and ID23-1 for CBC-ARS2. Data were integrated using the XDS suite[49] and analysed using the CCP4i suite[50]. Structures were solved by molecular replacement with PHASER[51] using the known structure of CBC bound to m7GpppG (PDB:1H2T). Crystals of m7GTP-CBC-ARS2 were of space group P1 with eight complexes in the asymmetric unit. Two of the unit cell angles were 90° (the other being 90.3°) and although the data integrate only marginally better in P1 than P2₁ ($R_{meas}$ 0.126 and 0.145, respectively) refinement is only possible in P1. Depending on the quality of the electron density, 11–14 residues from the ARS2 C-terminal peptide were placed in each of the eight CBC complexes and only B-factors were refined, not occupancies, which were set to 1. Crystals of m7GTP-CBC-NELF-E were of space group P2₁ with four complexes in the asymmetric unit. NELF-E peptide was placed in three of the four CBC complexes with 14–18 residues depending on the electron density quality. Refinement was performed with REFMAC5[52] or BUSTER (https://www.globalphasing.com/buster/) with local non-crystallographic symmetry restraints. Structure figures were drawn with PyMOL[53].

**Data availability**. Coordinates and structure factors for the CBC–ARS2 and CBC–NELF–E structures are available in the wwPDB with accession numbers 5OO6 and 5OOB, respectively. Other data are available from the corresponding author upon reasonable request.

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

## Acknowledgements

We thank the staff of the ESRF-EMBL Joint Structural Biology Group, in particular Matthew Bowler, for access to and help on ESRF beamlines. We thank the EMBL Grenoble Eukaryotic Expression Facility and high-throughput crystallisation facility (HTX), and the EMBL Heidelberg Proteomics Core Facility. We also thank Thomas Bock and Martin Beck (EMBL Heidelberg) for sharing their cross-linking protocols and Edouard Bertrand (CNRS, Montpellier) and colleagues for enlightening discussions. This work used the platforms of the Grenoble Instruct Center (ISBG: UMS 3518 CNRS-CEA-UJF-EMBL) within the Grenoble Partnership for Structural Biology (PSB), with support from FRISBI (ANR-10-INSB-05-02), GRAL (ANR-10-LABX-49-01).

## Author contributions

W.M.S. planned, performed, and analysed all biochemical and biophysical experiments. S.C. conceived and directed the project and performed crystallographic analysis. W.M.S. and S.C. wrote the manuscript.

## Additional information

**Competing interests:** The authors declare no competing financial interests.

