## [Peer Review file · Nature Communications]

Reviewers' comments:

Reviewer #1 (Remarks to the Author):

The manuscript by Schulze and Cusack addresses how the heterodimeric nuclear cap binding complex (CBC) engages multiple protein interaction partners, which ultimately dictates diverse gene expression outcomes ranging from export of short mRNAs and small-nuclear RNA (snRNA) to nuclear RNA turnover. Their study focuses on three protein interaction partners of CBC, including ARS2, NELF-E and PHAX, which play important roles in histone mRNA processing, transcription elongation and export of snRNAs, respectively.

The authors biochemically map interactions between ARS2 and NELF-E with CBC starting with full-length (or nearly full-length proteins) and proceeding with truncation analysis to identify minimal CBC interacting peptides. Peptides sufficient for binding reside at the C-terminus of NELF-E and ARS2. Isothermal titration calorimetry corroborates the peptide interactions. Likewise, a similar analysis is performed with PHAX. Interestingly, binding of ARS2 and NELF-E with CBC is enhanced 10-fold by m7GDP cap analog but much less so for PHAX (~ 2-fold), suggesting ARS2 and NELF-E have a similar binding mechanism, which is thermodynamically coupled to cap binding.

The authors next ask which complexes of CBC with ARS2, NELF-E and PHAX are compatible or mutually exclusive using competition assays. This includes classical cap chromatography with full-length (or nearly full-length proteins) or fluorescence polarization experiments (FP) with the aforementioned minimal CBC binding regions. The data suggest ARS2 and NELF-E bind CBC mutually exclusive whereas PHAX can form a higher order complex with CBC and ARS2. This finding indicates PHAX and ARS2 have different binding surfaces on CBC. In contrast, PHAX cannot bind CBC simultaneously with NELF-E, probably because regions outside the C-terminus, such as its RRM domain, bind CBC in a manner competitive with PHAX. In support, constructs containing only the C-terminal peptide of NELF-E can co-occupy CBC with PHAX, as indicated by competition FP experiments.

Crystallographic analyses reveal ARS2 and NELF-E engage CBC by molecular mimicry. The 2.8 Å structures contain cap analog or m7GDP, CBP20, CBP80 and the C-terminal peptide of ARS2 or NELF-E (residues 827-871 and 360-380, respectively). Peptide binding results in formation of an exquisite 3-way interface with CBP20 and CBP80. Conserved residues of the N- and C-termini of the peptides form bipartite interactions with proximal and distal pockets of the CBP20/80 heterodimer. The structure is well-supported by mutagenesis applied in conjunction with quantitative ITC binding assays. These experiments are properly controlled, by cap pull-down chromatography with CBC to evaluate if mutants result in a general loss of CBC. Because high-resolution structural studies of PHAX with CBC have been elusive, the authors evaluate contacts by chemical cross-linking in conjunction with mass-spectrometry analysis. The data suggest a different binding site for PHAX than C-terminal peptides of ARS2 and NELF-E. This result, together with the observation that PHAX and full-length NELF-E compete for CBC, may provide clues to how additional domain contact the CBC.

An interesting feature of the structural studies is that residues that normally contact each other in the CBC20/80 heterodimeric interface are repositioned by rotameric changes to accommodate binding of ARS2 and NELF-E C-terminal peptides. Analysis of previously determined structures of CBC in the presence and absence of cap compared to new structures with ARS2 or NELF-E peptides reported in the manuscript reveal that residues in the CBP20/80 interface, such as K77 of CBP20 are poorly positioned for interactions with the peptides in the absence of cap. Thus, the thermodynamic coupling between cap recognition and peptide binding by CBC is born out by the new crystal structures. Intriguingly, conserved tyrosine residues of ARS2 and NELF-E within the proximal binding site are known to be phosphorylated, so the structural studies suggest binding of ARS2 and NELF-E to CBC could be regulated by phosphorylation.

Overall, the work is a biochemical and structural tour de force. The authors suggest that NELF-E and ARS2 could function in different steps of a common pathway by binding CBC. For example, NELF-E binding paused after the transcription of the first 20-30nt by POLII. It could then get displaced by ARS2 and PHAX to promote export of U snRNA which is critical for splicing. Alternatively, but not mutually exclusive, ARS2 could act engage CBC and adaptors for the NEXT or PAXT complex to promote 3'-5' nuclear decay by the exosome. The structural studies in conjunction with prior phosphoproteomic analyses suggest interactions of ARS2 and NELF-E with CBC may be antagonized by tyrosine phosphorylation. Though the functional significance of this study awaits validation in cells, clearly publication of this manuscript will stimulate additional research in the field. Accordingly, I recommend publication in Nature Communication after the following issues are addressed.

Major comments:

1-A prediction of the authors' model in Fig S4 is that the 10-fold increase in peptide binding affinity observed for the CBC complex in presence of cap would be reduced by mutations in the K77 loop of CBP20 (Fig S4).

2-Related, does NELF-E peptide binding increase the cap affinity of CBC? For example, enhanced affinity of CBC for cap could be important to ensure that CBC stays on the cap when in complex with NELF, which could prevent precocious nuclear decapping and subsequent RNA decay.

3-The manuscript requires some reorganization, because in its current form it is very difficult to read. For example, the introduction is very long and dense, containing many protein names and acronyms that could be confusing for a general audience. Since the manuscript is about interactions CBC with ARS2, NELF and PHAX, perhaps these proteins and their biology should be emphasized in the introduction and implications for additional complexes such NEXT and PAXT moved to the discussion.

Minor comments:

4-The binding surface of PHAX revealed by cross-linking MS should be compared to the structural data of CBC with ARS2 and NELFE and discussed.

5-The crystallographic methods are somewhat sparse, and it would be useful to know how the partial occupancy of peptides was treated in the solution and refinement.

6-The protein labels in Figs 4 C-H are too small: this is important since different constructs for NELF-E (the C-terminal peptide or that containing containing flanking N-terminal RRM) are employed. The small labels are illegible and make this crucial difference very difficult to recognize. These data were very confusing at first glance.

7-In Figure 5B, left panel, the labels for ASP107 and GLN55 of CBP20 are swapped.

8-In Figure 5C, the ITC data indicating the F871D mutation (in the context of the triple mutation) reduce binding of ARS2 are rather marginal, supported only by a small difference in heat evolved. Could this be corroborated by a different method?

9-In Figure 6, again the labels for Asp107 and GLN55 of CBP 20 are swapped; see comment 3 above. In addition, there are a number of errors in the labeling of amino acids in Fig 6C: Tyr854 of ARS2 should be 859; Phe380 of ARS2 should be 871.

Reviewer #2 (Remarks to the Author):

The authors present the structural basis for mutually exclusive interactions between the nuclear cap complex, ARS2, and NELF-E. The nuclear cap complex (CBC) not only protects nascent RNAPII transcripts from degradation, but acts as a scaffold for the recruitment of processing, termination, transport, export, and degradation complexes in the life of the half dozen or so different types of transcripts generated by RNAPII. The CBC has been shown to mediate the various functions through the formation of mutually exclusive complexes, although very little is known mechanistically about how this is achieved. Ars2 is a critical to CBC function in many of these interactions. NELF-E is part of the negative elongation factor complex and causes RNAPII pausing shortly after transcription initiation that is thought to be important for CBC recruitment to the m7G-cap. In addition, NELF-E participates in histone mRNA 3' end processing. PHAX has been shown to form a quaternary structure with the CBC and Ars2 and is necessary for snRNA export. In this study the authors elegantly map the interaction regions between Ars2, NELF-E and the CBC and they show by chemical crosslinking putative interaction regions between PHAX and the CBC. Using biochemical and biophysical approaches, the authors show that Ars2, and PHAX interaction with CBC is mutually exclusive of NELF-E. They then go on to show through co-crystals, the basis for the exclusivity, which is that NELF-E and Ars2 bind to the identical site on an interface between CBP20 and CBP80. They identify a previously unappreciated conservation between the NELF-E and Ars2 binding peptides. Finally they support their co-crystals through mutagenesis studies of the Ars2, Nelf-e, and CBP20/80 binding interfaces. On the whole, I found this manuscript to be very well composed and quite compelling. There is high difficulty here, as I'm aware of others who have tried unsuccessfully to obtain co-structures of these molecules. I think the data generated will be of interest to the wide readership of Nature Communications. The data generated is highly novel and well supported. My one comment/suggest is:

1) Phax, Ars2, and CBC form a ternary complex. The authors present preliminary crosslinking data identifying the putative PHAX interaction site on CBC. It would be helpful if the authors identified the Phax crosslinking sites on CBC subunits on the structure of the CBCA either on the existing figure or if necessary an additional view. Similarly, based on their data, binding of NELF-E or Phax is also mutually exclusive. Inclusion of the Phax crosslinking would be helpful for future work. It is intriguing that the inclusion of 244-354 in NELF-E which contains the RRM domain was necessary to block PHAX interaction with CBC, which they suggest is due to further interactions between NELF-E and CBC. Are the distances between the putative site on the CBC and the NELF-E binding site compatible with this model?

Reviewer #3 (Remarks to the Author):

In their manuscript entitled "structural basis for mutually exclusive co-transcriptional nuclear cap-binding complexes with either NELF-E or ARS2" Schulze & Cusack characterize interactions between CBC and three partner proteins, NELF-E, ARS2 and PHAX by various biochemical and biophysical methods. Here, the authors show that the homologous C-terminal peptides of NELF-E and ARS2 bind identically to the CBP20-CBP80 interface and demonstrate that two mutually exclusive complexes CBC-NELF-E and CBC-ARS2-PHAX are formed.

They additionally solve the crystal structure of the m7GTP-1 CBC-ARS2827-871 complex. While the reviewer is unable to judge the quality of the crystallization work and the thus resulting structure, the work appears to be thoroughly done and the conclusions justified.

In order to address CBC and PHAX interactions - as crystallization attempts have failed - the authors use cross linking coupled to mass spectrometry (XL-MS) in order to define protein-protein contact sites.

For XL-MS the authors use the xQuest/xProphet software suite, which arguably is the standard in the field.

Indicated settings for the xQuest/xProphet pipeline are reasonable. In particular, while the

authors do not report a false-discovery-rate (FDR) for identified cross-links, they use an Id score of >28, which seems a reasonable assumption given the three protein complex under investigation.

The XL-MS data therefore appears to be thoroughly carried out and well done.

Minor Comments:

Throughout the text: "mass spectroscopy" should be replaced by the term "mass spectrometry" (MS)

And

"cross-linking MS" by the term "chemical crosslinking coupled to MS (XL-MS)".

Supplementary Table 1:

The caption should read – "Inter" protein cross-links, as the links between different polypeptide chains are shown in the table.

Also, as now false-discovery-rate is shown, the term "xProphet" should be deleted.
e.g. "The Id (linear discriminant) confidence scores were calculated by xQuest"

The Id score should indicate the whole range – e.g. Id of >28 was used.

Finally, as the authors show also Intralinks in Figure 3B, those links should also be shown in a Supplementary Table or added to the current Table 1.

Replies to the reviewers' comments:

Reviewer #1:

Major comments:

1-A prediction of the authors' model in Fig S4 is that the 10-fold increase in peptide binding affinity observed for the CBC complex in presence of cap would be reduced by mutations in the K77 loop of CBP20 (Fig S4).

This and the following point concern the mechanism for the increased affinity of the ARS2 or NELF-E peptide for m⁷GTP-bound CBC rather than for CBC alone and vice versa. As mentioned in the text the mechanism is not obvious from the structures although superposition by means of CBP80 of the apo (PDB:1H2V), cap-bound (PDB:1H2T) and the ARS2 (or NELF-E)-CBC-m⁷GTP complexes suggests that m⁷GTP binding induces a slight displacement of CBP20 relative to CBP80 bringing it closer to the peptide bound conformation. This is particularly apparent for the conformation of the CBP20 K77-loop and the Y50-loop (Fig. S4) but is probably a more global effect resulting from the induced fit structuring of much of CBP20 upon cap-binding. We therefore do not think that it is straightforward to decouple this synergistic effect in a simple or meaningful way by mutation. Most illuminating, but seemingly unattainable due to the low affinity, would probably be a structure of the CBC with just the bound peptide.

2-Related, does NELF-E peptide binding increase the cap affinity of CBC? For example, enhanced affinity of CBC for cap could be important to ensure that CBC stays on the cap when in complex with NELF-E, which could prevent precocious nuclear decapping and subsequent RNA decay.

This is a good suggestion which we have now tested. Upon saturating CBC with NELF-E or ARS2 C-terminal regions and then titrating with m⁷GTP we find by ITC that the affinity for cap is indeed increased by a factor of ~3 for pre-bound NELF-E and ~7 for pre-bound ARS2 (new Supp. Fig. 7, referred to on Page 16, line 9-12; K_D values added to the bottom of Table 1). Again this could be explained by peptide binding inducing a displacement of CBP20 that then favours cap-binding (perhaps by partially ordering CBP20?), but a much more detailed study (e.g. by NMR or a structure of CBC with only bound peptide) would be needed to understand the mechanism in detail. The fact that NELF-E binding to CBC enhances cap-binding is also now mentioned in connection with the recruitment by NELF-E of CBC to the newly capped transcript (Page 18, lines 9-12).

3-The manuscript requires some reorganization, because in its current form it is very difficult to read. For example, the introduction is very long and dense, containing many protein names and acronyms that could be confusing for a general audience. Since the manuscript is about interactions CBC with ARS2, NELF and PHAX, perhaps these proteins and their biology should be emphasized in the introduction and implications for additional complexes such NEXT and PAXT moved to the discussion.

Following the referees suggestion, we have simplified the introduction (979 words now instead 1098) and moved mention of NEXT and PAXT to the discussion.

Minor comments:

4-The binding surface of PHAX revealed by cross-linking MS should be compared to the structural data of CBC with ARS2 and NELFE and discussed.

In new Supp. Fig. 7, we display the PHAX cross-links on CBC on the structure of m⁷GTP-CBC-ARS2. This shows that the putative PHAX binding site is in relatively close proximity to the ARS2/NELF-E binding site, making steric competition between PHAX and NELF-E RRM plausible.

5-The crystallographic methods are somewhat sparse, and it would be useful to know how the partial occupancy of peptides was treated in the solution and refinement.

We added the following to the methods section on structure determination:

‘Depending on the quality of the electron density, 11-14 residues from the ARS2 C-terminal peptide were placed in each of the eight CBC complexes and only B-factors were refined, not occupancies (which were set to 1.0). Crystals of m⁷GTP-CBC-NELF-E were of space group P21 with four complexes in the asymmetric unit. NELF-E peptide was placed in three of the four CBC complexes with 14-18 residues depending on the electron density quality.’

6-The protein labels in Figs 4 C-H are too small: this is important since different constructs for NELF-E (the C-terminal peptide or that containing flanking N-terminal RRM) are employed. The small labels are illegible and make this crucial difference very difficult to recognize. These data were very confusing at first glance.

We have improved the clarity of this figure by increasing size of labelling, representing the FAM fluorophore as a star and representing the different ARS2 and NELF-E constructs used by slightly different shapes. Moreover the latter points are now highlighted in the figure legend.

7-In Figure 5B, left panel, the labels for ASP107 and GLN55 of CBP20 are swapped.

Corrected

8-In Figure 5C, the ITC data indicating the F871D mutation (in the context of the triple mutation) reduce binding of ARS2 are rather marginal, supported only by a small difference in heat evolved. Could this be corroborated by a different method?

We agree that since the double mutant essentially does not bind in the ITC experiments, it is not possible to claim that addition of the F871D mutant further reduces binding. We therefore prefer to leave out the data for the triple mutant.

9-In Figure 6, again the labels for Asp107 and GLN55 of CBP 20 are swapped; see comment 3 above. In addition, there are a number of errors in the labeling of amino acids in Fig 6C: Tyr854 of ARS2 should be 859; Phe380 of ARS2 should be 871.

Corrected

Reviewer #2:

The data generated is highly novel and well supported. My one comment/suggest is:

1) Phax, Ars2, and CBC form a ternary complex. The authors present preliminary crosslinking data identifying the putative PHAX interaction site on CBC. It would be helpful if the authors identified the Phax crosslinking sites on CBC subunits on the structure of the CBCA either on the existing figure or if necessary an additional view. Similarly, based on their data, binding of NELF-E or Phax is also mutually exclusive. Inclusion of the Phax crosslinking would be helpful for future work. It is intriguing that the inclusion of 244-354 in NELF-E which contains the RRM domain was necessary to block PHAX interaction with CBC, which they suggest is due to further interactions between NELF-E and CBC. Are the distances between the putative site on the CBC and the NELF-E binding site compatible with this model?

This is a very similar comment to that made in minor point 4 of referee 1. In new Supp. Fig. 7, we display the PHAX cross-links on CBC on the structure of m⁷GTP-CBC-ARS2. This shows that the putative PHAX binding site is in relatively close proximity to the ARS2/NELF-E binding site, making steric competition between PHAX and NELF-E RRM plausible.

Reviewer #3:

The XL-MS data therefore appears to be thoroughly carried out and well done.

Minor Comments:

Throughout the text: “mass spectroscopy” should be replaced by the term “mass spectrometry” (MS). And “cross-linking MS” by the term “chemical crosslinking coupled to MS (XL-MS)”.

These changes have been done.

Supplementary Table 1: The caption should read – “Inter” protein cross-links, as the links between different polypeptide chains are shown in the table. Also, as now false-discovery-rate is shown, the term “xProphet” should be deleted. e.g. “The ld (linear discriminant) confidence scores were calculated by xQuest”

The ld score should indicate the whole range – e.g. ld of >28 was used.

Finally, as the authors show also Intralinks in Figure 3B, those links should also be shown in a Supplementary Table or added to the current Table 1.

Supp. Table 1 now includes both inter- and intra-subunit cross-links. The other suggested changes have also been implemented.

REVIEWERS' COMMENTS:

Reviewer #1 (Remarks to the Author):

The revised manuscript addresses my concerns, and I think those of the other referees as well. It is now suitable for publication in Nature Communications.